# Adopting Augmented Reality to Engage Higher Education Students in a Museum University Collection: the Experience at Roma Tre University

**Antonella Poce \*, Francesca Amenduni, Carlo De Medio, Mara Valente and Maria Rosaria Re**

Department of Education, Roma Tre University, Roma 00185, Italy; franesca.amenduni@uniroma3.it (F.A.); carlo.demedio@uniroma3.it (C.D.M.); mara.valente@uniroma3.it (M.V.); mariarosaria.re@uniroma3.it (M.R.R.)

\* Correspondence: antonella.poce@uniroma3.it

**Abstract:** University museums are powerful resource centres in higher education. In this context, the adoption of digital technologies can support personalised learning experience within the university museum. The aim of the present contribution is to present a case study carried out at the Department of Educational Sciences at Roma Tre University with a group of 14 master's degree students. Students were involved in a 2-h workshop in which they were invited to test augmented reality technology through a web app for Android. At the end of the visit participants were required to fill in a questionnaire with both open-ended and closed-ended questions aimed at investigating their ideas on the exhibition and their critical thinking level. Students appreciated the exhibition, especially its multimodality. Most of the frequent themes identified in open-ended answers are related to critical and visual thinking. Despite the positive overall evaluation, there is still room for improvement, both in terms of technology and educational design.

**Keywords:** augmented reality; university museum; critical thinking

## 1. Introduction

University museums are powerful resource centres in higher education to enhance the impact of their teaching and research [1]. Today, university museums and collections are an essential part of a country's heritage and are becoming more accessible to a wider public [2]. They provide unique opportunities to actively fulfil critical mission statements of university, by inspiring innovative research projects and providing students with real life objects [3] to explore and manipulate. Museum collections within the university setting offer a valuable opportunity to provide subject specific knowledge in various domains, from STEM education to arts and humanities. Museum objects can also be used to inspire discussion, group work and critical thinking—all crucial and transferable key skills in higher education. University collections comprise a wealth of information, documenting the historic rise of disciplines and representing cultural and natural diversity from across the globe [4].

However, some studies have highlighted that university museums are facing similar struggles across departments and fields: lack of financial resources, time and support [5]. Researchers, museum curators and higher educational staff felt the need to understand how university museums could make visible their institutional relevance for universities and for society in general. For these reasons, there has been an increase in the number of initiatives to enhance research and organisms of collaboration and cooperation in the field of university museums. One of those initiatives was the establishment of the University Museums and Collections Committee (UMAC), one of the independent committees of the International Council of Museums (ICOM) in 2001 [6]. UMAC provides a trustworthy and international perspective regarding best practices and successful experiences within the context of

university museums. Recent reports promoted by UMAC highlighted the role of digital and emergent technologies to further develop the presentation of museum collections through digital media to an ever-widening audience [7].

The adoption of digital technologies to support a personalized learning experience within museum contexts is a prominent trend in recent years. Museums can provide additional information through mobile apps, improving curiosity and engagement towards exhibitions and collections. As QR codes and smartphone apps have become widely used in museums all over the world, some museums are starting to explore ways to enhance the visitor experience through more interactive and customized devices [8]. Specifically, augmented reality has become an accessible tool for promoting the interaction with real-world objects and pieces of arts included in museum collections. Different studies have highlighted that adopting augmented reality solutions in museum contexts can support the visitors' overall experience [9], their intention to come back to the museum, their purchase intention [10] and a critical understanding of challenging knowledge [11].

Educational policy makers identify critical thinking as an essential driver for progress and knowledge growth in any field and in the broad society. For example, the World Economic Forum [12] states that critical thinking will be the second most important skill in 2020 to face the growth of automatization. Critical thinking is commonly defined as 'purposeful, self-regulatory judgment that results in interpretation, analysis, evaluation, and inference, as well as explanations of the considerations on which that judgment is based' according to the definition produced in the Delphi Report [13]. Experts agreed with the idea that critical thinking can be considered both in terms of skills (such as evaluation, analysis, argumentation, inference) and dispositions (such as scepticism, open-mindedness, inquisitiveness). In recent years a growing number of research articles have been published with the purpose of investigating pedagogical practices that can support Critical Thinking development. Some of the strategies that have been recognised to promote critical thinking are reflective questioning [14,15] and visual thinking [16,17]. According to Housen, watching and enjoying a work of art, comparing different artworks, artists, styles, and trends, allows the museum audience to develop their thinking, analysis and evaluation skills. The use of digital tools in the field of arts and cultural heritage education represents a real innovation challenge because new teaching and learning methodologies may be developed. Indeed, during the last decades, different pedagogical approaches that combine the use of technologies with cultural heritage education have been developed, such as object based learning [18], visual thinking [19], and digital story telling [20]. In addition, it should be also considered how the progress made about the web and the related technologies, affects cultural heritage educational practices. It was shown that different affordances of the Web are related with different cognitive processes that can be enabled [21,22] and, as a consequence, different transversal skills. The Centre for Museum Studies, based at Roma Tre University, has studied for several years the relation among cultural heritage education, technology and critical thinking both in national and international experimentation. This experimentation is situated in the intersection among these educational practices and competences.

Considering the above-mentioned premises, the goal of the research herewith presented was to assess the impact of an augmented-reality experience through a mobile phone application on students' development, especially in terms of critical thinking, in the context of a university museum.

## 2. The Context of the Research

The aim of the present contribution is to introduce the experience carried out at the Roma Tre University Department of Education with a group of 14 master's degree students. The Department of Educational Sciences hosts a permanent exhibition of paintings realized by a contemporary Italian artist named 'Tito Rossini'. Students involved in the experimentation attended a university module named 'Experimentalism, museum and reading'. The course has the general purpose of illustrating the main research methodology guidelines in the field of cultural heritage education. The specific objectives addressed by the module are the following:

- Acquiring theoretical and methodological foundations of the experimental investigation in the field of museum education and cultural mediation;
- Defining a research problem in the context of the use of artistic and cultural heritage;
- Formulating a research hypothesis;
- Identifying the main data collection and evaluation tools for museum education courses;
- Knowing examples of good research practices in the field of artistic and cultural heritage education.

In order to achieve these learning goals, it was decided to involve university students in an interactive learning path designed within the Tito Rossini exhibition. More specifically, the activity was thought to give students the opportunity to acquire knowledge regarding research practices in the field of cultural heritage education and to reflect upon the role of university museums.

The Tito Rossini permanent exhibition is located at the Roma Tre University Department of Education, in the centre of Rome. University teaching staff rooms, administrative staff offices, the library staff offices, and the reception room of the library itself are located in the building. The Rossini permanent collection was donated to the University by the artist in 2018. At the beginning, the exhibition did not provide cultural mediation tools, except for small labels containing basic information (name of the author, the title of the painting and the date of production).

In order to offer a new receptive approach of the collection itself, new mediation tools have been designed and realized within a project named *Inclusive memory*. The Centre for Museum Studies (CDM) of the Department of Education realized a set of innovative mediation tools aimed at supporting the visitors' learning experience within the exhibition.

Mediation tools created for the exhibition are heterogeneous and multimodal. In the first phase of the project, the following tools, expected to provide different kinds of learning paths within the visit, were introduced:

- A description of the work of art from an art history point of view;
- An audio description of the work of art linked to a QR code;
- A short story acted, recorded and stored in the QR code;
- A music track.

In a previous research [23], a positive overall evaluation of the innovative mediation tools realized within the Tito Rossini project was shown.

In the second phase of the project, an augmented reality system was set up to engage participants and it is the core content of the present paper.

## 3. Methods

As mentioned above, the experimental activity held at the Department of Education-Roma Tre University, based on a multimodal use of the contemporary art permanent exhibition *Tito Rossini*, involved 14 students (average age = 25, 14; standard deviation = 3, 53; female = 12; gender not specified = 2). A small group of participants was involved because the Augmented Reality fruition modality was tested for the first time in the Tito Rossini exhibition and it was necessary to monitor and support the group of students during the visit.

Time devoted to the experience was two hours in total. In the initial phase, a brief introduction on the artist 's biographical notes and a focus on the themes developed within his works were presented. Tito Rossini's work was contextualized in the broader Italian art scene of the early '900, by referring to important predecessors that inspired the entire work of the artist from Formia: Morandi, Carrà, Trombadori, Oppi. The design of the visit was organized with the aim of involving students directly as active subjects in the fruition of the works. The visit involved a *gamified* use of the works; through a web app the students were immersed in a virtual reality experience where, moving from one painting to another, they were asked to interact by answering to the questions appearing on the screen of their smartphones, which adapted their contents according to the subject framed. Positive advancement

in the game generated a final question that stimulated the students to reflect critically on the entire exhibition path.

The questions included in the app were designed with the intention to investigate, in the first instance, each participant's history of art knowledge level (Table 1): from the acquaintance with 'primary colours', to critical reflection starting from direct investigation of the picture, to observation of the work of art, to the assessment of artistic techniques knowledge.

**Table 1.** Questions related to each of Tito Rossini's paintings.

| Tito Rossini's Paintings | Questions |
|---|---|
| Terrazzo con panni stesi<br>*Balcony with hanging clothes* | 1. From what can you assume that the scene depicted is set in the morning?<br>- From the length of the shadows<br>- From the colour of the sky<br>- From the hanging clothes |
| Una partita di bocce<br>*Bowles game* | 2. Look at the balls. Can you recognize primary colours?<br>- White and red<br>- Red<br>- Black |
| Nel giorno di festa<br>*On a celebration day* | 3. It seems that someone has just finished his breakfast. Do you remember the name of an artist who painted a famous breakfast?<br>- Monet<br>- Manet<br>- Pisarro |
| Natura morta con porta bianca<br>*Still life with white door* | 4. Do you see a living element in the painting?<br>- Olive branch<br>- Jar<br>- None |
| Una sola rotta<br>*The only route* | 5. Look at the painting. Can you recognize the painting technique?<br>- Panel painting<br>- Oil on canvas<br>- Watercolour on canvas |

The last and sixth question was designed to stimulate the links between themes and artists in art history 'After visiting the exhibition, reflect on what you have observed: there is a recurring element in many of the exhibited works, can you tell what it is?'. The survey integrated in the augmented reality tour app led visitors to stimulate the critical view of the experience of the entire collection. The focus here in fact was on the linking narrative of the paintings exhibited. Rossini's intimate vision, lifestyle and landscapes are episodes of everyday life that all together tell the story of the artist. It is precisely the element of wire, declined in different variations, that keeps together realistically and ideally works that are quite different from each other.

The invitation to reflect on the *trait d'union* among the different works has been specifically projected to stimulate and activate each participant's critical thinking skills. Participants were in fact prompted to focus not only on what has been observed but also on the message, the 'reason' behind each work, their meaning as individual works of art and their role in the entire exhibition system.

Each painting tells a fragment, a snapshot of the painter's everyday life. The combined set of works tell the story of the painter's life, express his childhood memories, the emotions and lightness of a summer night, the sea breeze and the warm August sun, the scent of clothes lying outside and the walks through the narrow-cobbled streets. The image becomes a story and allows an attentive observer to interact with the painter's soul.

*3.1. The Web-App for Augmented Reality*

The web-app developed is an Android application created within the Unity3D graphics engine (https://unity3d.com/unity). Unity is a multiplatform graphic engine developed by Unity Technologies widely used in the field of video games, virtual and augmented reality applications. Unity allows you to create applications using one of the three programming languages (C #, javascript, Boo) and

to export them to various devices, from the PC or Mac to the Android and IOS mobile devices. This graphic engine has been chosen because it allows communication with the Vuforia platform via free plug-ins (https://developer.vuforia.com/). Vuforia is a platform that deals with extracting features that characterize objects and creating databases of recognizable models for augmented reality. The features associated with the images are a set of information about the image (often a set of characterizing points extracted through different algorithms) that allow to recognize the object by analysing a video stream.

The paintings selected for the experimentation were digitalized in high definition and processed in order to identify the main features of each work in the collection. The extracted features were imported into the graphic engine. Then, a virtual 3D assistant, named Mila, was created in order to ask questions to users regarding the selected paintings.

The animation of 3D models is a complex task. For this reason, our virtual assistant movements were implemented by the use of a free service, named Mixamo (https://www.mixamo.com/). The service provides a set of animations ready and usable for any 3D model. However, the main constraint is that the 3D model should have a humanoid shape, which can be identified by some basic elements of the body such as head, a bust, legs and arms.

The android application can recognise, through the mobile device camera, a piece of art uploaded in the database. When the application recognizes the painting, a digital version of the work of art appears on the screen together with the virtual assistant who propose a question for the user. Users can provide answers to a multiple-choice question and they can try to find the correct answer as many times as they want. This because the questions of the quiz were not aimed at providing a summative assessment for the teacher, but at enhancing participants' interaction with the painting and engagement during the visit. In the case of a wrong answer, the assistant will mirror the visitor's emotions by showing her disappointment. On the other hand, after the selection of the correct answer, the assistant starts celebrates dancing and the user will earn a point. For each correct answer, users can also visualize a part of the final question. The left part of Figure 1a shows what the users see if they correctly answer to the question regarding the third painting. It appears not only a picture of the painting, but also a piece of the final question, above the picture.

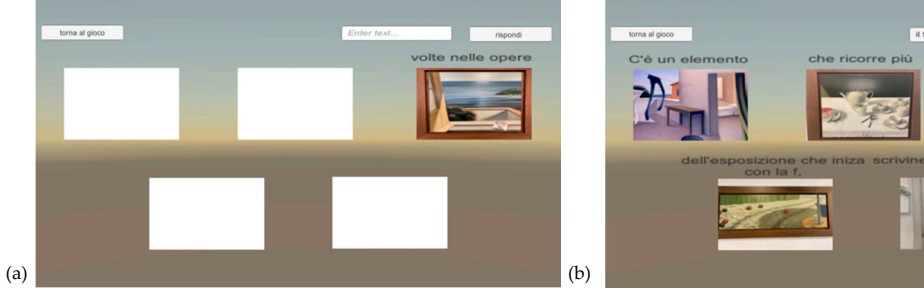

**Figure 1.** Overview of the web-app. The picture (**a**) shows what the users see if they correctly answer to the question regarding one of the paintings. The picture (**b**) shows what the users see if they correctly answer all the questions.

By correctly answering all the questions, the user will eventually be able to read the entire final question as shown in the right part (b) of Figure 1. The last question regards the *file rouge* that links all the works of art presented in the collection. By correctly answering also to the final question, the app will present a message of victory and invite users to the next experience.

### 3.2. Data Collection and Data Analysis

At the end of the visit, students were asked to fill in a questionnaire created through Google modules. The questionnaire was filled in anonymously in order to respect the participants' privacy

and to not inhibit participants to also providing negative feedback regarding the exhibition. The questionnaire was made out of three sections:

- In the first section there were three personal questions: gender, age and occupation;
- The second section was devoted to asking questions regarding students' previous knowledge about the Tito Rossini exhibition, the evaluation of their learning experience, and suggestions for improving the learning path. In this section, both closed-ended questions and open-ended questions were included. For the questions regarding the exhibition's assessment, students expressed their opinion trhough a Likert scale from 1 (totally disagree) to 5 (totally agree). In addition, participants were asked to describe the multi-modal exhibition by the use of a metaphor.
- In the last section students were invited to answer to a last open-ended question aimed at assessing their critical thinking skills level. The question was 'From your point of view, what is the role of university museums in the wider city museum system? What function do they have for the territory and what for the University? What role can technology play in university museums?'

Different methods of data analysis were applied. average scores were calculated on quantitative data collected through closed-ended questions. A theme analysis was applied to the open-ended-questions presented in the second section. Theme analysis is related to the classification of the patterns presented from qualitative data into themes. Themes can be defined as units derived from patterns such as topics, recurring activities, meanings and feelings. Through this technique, it is possible to bring together components or fragments of ideas or experiences, which often are meaningless when taken alone [24]. We then calculated the percentage of occurrence (O) of the defined themese into the open-ended questions. In addition, we calculated the the percentage of occurrence of the themes most commonly associated with positive and negative comments regarding the exhibition. The software Atlas.ti, one of the most popular softwares to carry out qualitative analysis [25], was used for theme analysis. Finally, the final open ended question was treated with content analysis to assess critical thinking [26] levels. The critical thinking assessment model adopted is based on six macro-indicators: use of language, argumentation, relevance, importance, critical evaluation and novelty. Each macro-indicator can be assessed with a minimum score of 1 to a maximum score of 5. Thus, the maximum score possible is 30.

## 4. Results

The group of participants was composed by 14 master's degree students in educational sciences (Table 2). Nine out of fourteen students (64%) were part time students (employed students) whilst the remaining five were not involved in any professional activity at the moment of the experimentation.

**Table 2.** Sample characteristics.

| Participants | Number Involved |
|---|---|
| Status | |
| - Employed students | 9 |
| - Full time students | 5 |
| Gender | |
| - Female | 12 |
| - Not specified | 2 |
| Age | |
| - Between 21 and 23 | 7 |
| - Between 24 and 27 | 3 |
| - Between 28 and 30 | 4 |

Ten students had never heard about the artist Tito Rossini and only four knew something about him (Figure 2a). In addition, eight students had noticed the exhibition in the university building before the visit, whilst six had never noticed the exhibition before (Figure 2b). The following preliminary results show that the exhibition is not enough exploited and promoted at departmental level.

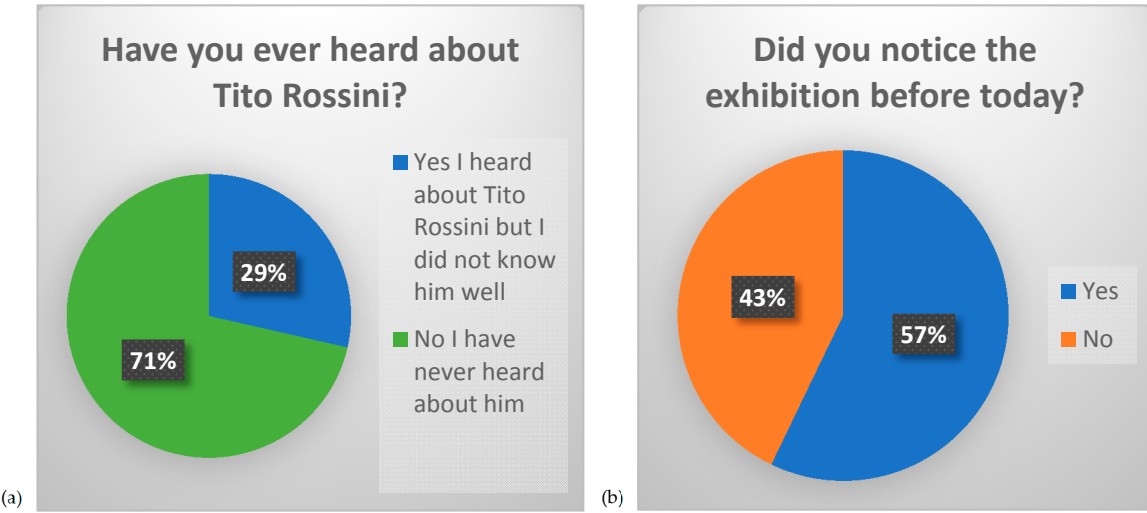

(a)　　　　　　　　　　　　　　　　　　　　　　　　(b)

**Figure 2.** Participants' knowledge about the artist Tito Rossini (**a**) and the exhibition (**b**).

Participants were asked to assess their level of agreement with two negative statements and eight positive statements regarding the exhibition on a Likert scale from 1 (totally disagree) to 5 (totally agree). It is possible to see that the average scores for the two negative statements is lower than 3 (Figure 3). Thus, in average, participants did not prefer the traditional visit (average = 2.78) compared to the multi-modal exhibition, and technology was not a distraction for them (average = 2.86).

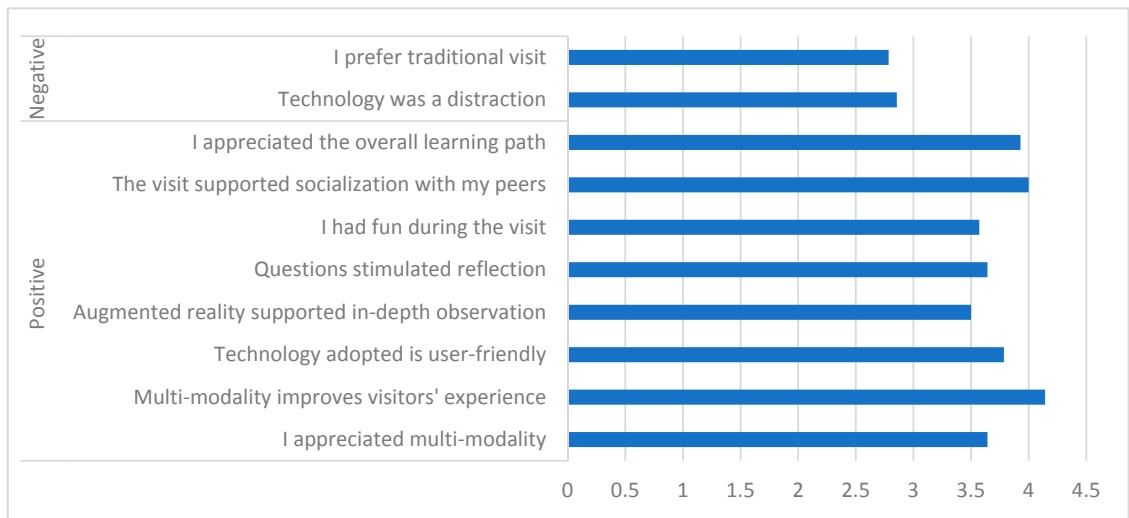

**Figure 3.** Participants' level of agreement with positive and negative statements regarding the exhibition. The level of agreement was expressed on a Likert Scale from 1 (totally disagree) to 5 (totally agree).

On the other hand, regarding positive statements the average score was always higher than 3.5 which indicates a positive attitude towards different aspects of the exhibition. In particular, the highest average score was obtained for the statements 'Multi-modality improves visitors' experience' (average = 4.14) and the 'The visit supported socialization with my peers' (average = 4). Despite the general appreciation, the average is lower for the statements 'I had fun during the visit' (average = 3.57) and

'Augmented reality supported in-depth observation of paintings' (average = 3.5). These results suggest some room for improvement of the next experimentation.

From the theme analysis, the following topics emerged (Figure 4): *inquisitiveness* (O = 25.93%) and *synthesis* (O = 7.41%), both connected to Critical Thinking skills and dispositions. *Art* (14.81%), *visualization* (O = 16.679) and *innovation* (O = 1.85%) more connected to the definition of Visual thinking.

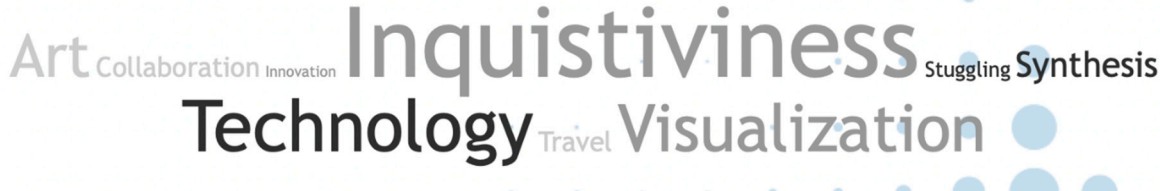

**Figure 4.** Emergent topics in participants' open-ended answers.

A co-occurrence between topics related with Critical Thinking and Visual Thinking was found. More specifically, it was found that *visualization* and *inquisitiveness* co-occur in the 35% of the cases. This relation was well shown in a metaphor adopted by one of the participants: '*the multi-modal interactive exhibition was like a magnifying glass*'.

Other topics, which emerged from participants answers, are *technology* (O = 18.52%), *collaboration* (O = 5.56%), *travel* (O = 5.56%), and *struggling* (O = 3.7%). The topic of *travel* was associated with the topic of *art* through a meaningful metaphor: "*The exhibition was a journey through painted window*".

In Figure 5 the positive and negative aspects of the Tito Rossini's exhibition according to participants are presented. The most appreciated features of the exhibition were the opportunity to identify a *Fil Rouge* among the paintings and to be stimulated by questions designed to support reflection. *Technology* was considered in both positive and critical aspects. Specifically, people with an iOS system would like to have the opportunity to install the app on their mobile phone. However, the experimentation was run with a limited amount of resources, thus it was not possible to design the App also for iOS system at this stage of the project. Participants would also like to have the time to focus also on the other paintings and they did not always appreciate the arrangement of the paintings in the building.

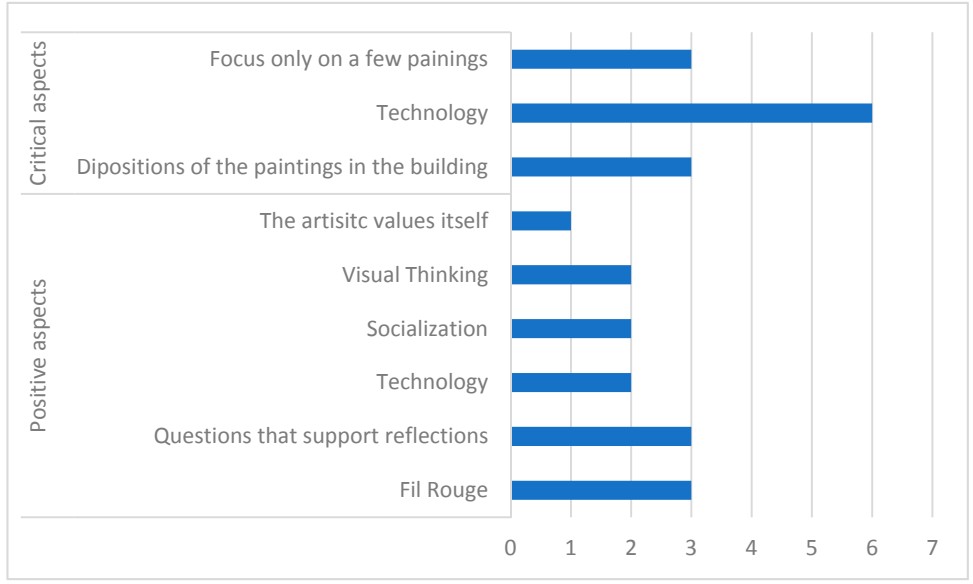

**Figure 5.** Positive and negative aspects of the exhibition according to the participants.

At the end, 10 out 14 students answered to the last open question aimed at assessing participants'
critical thinking level, asking them to reflect upon the role of university museums (Figure 6). All
the students obtained a score higher than 18/30 which indicates a sufficient level of critical thinking
( In the Italian system, 18 corresponds to the minimum score to pass a University exam http:
//attiministeriali.miur.it/media/240734/allegato_5.pdf). Only three students obtained a score higher
than 25/30 which indicates a high level of critical thinking. Below one of the students' comment
is reported:

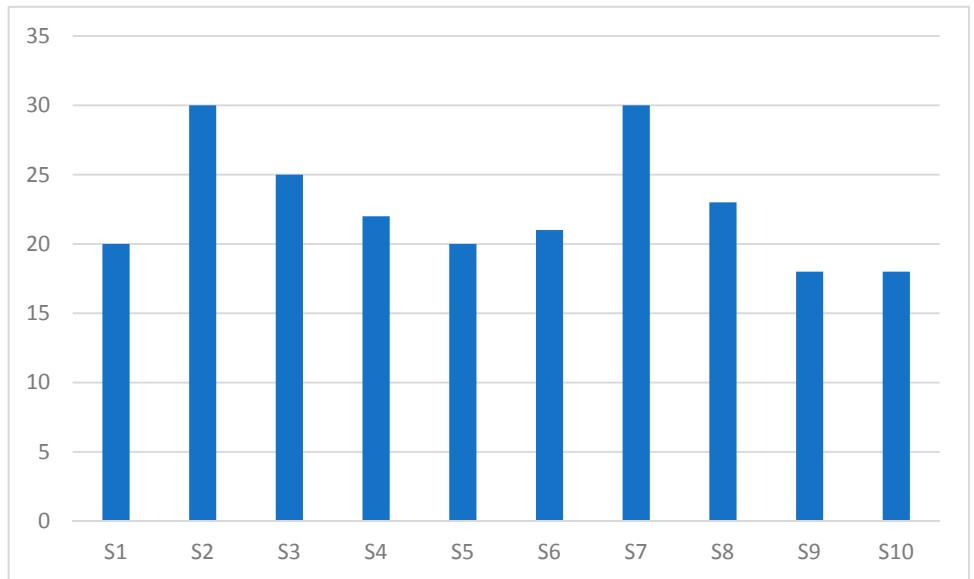

**Figure 6.** Critical thinking level shown by the participants to the last question of the questionnaire.

*'Universities offer numerous works of art that are not often valued adequately. It would be an opportunity
to bring the students' attention to them, perhaps even by teaching "on-conventional" lectures, not necessarily
connected to museum education: to overcome the face to face teaching and learning from academic standpoint and
carry out peer to peer teaching among the works of art that surround us. I believe that if developed to the fullest,
technology can be an excellent stimulus to encourage visitors to better observe the works and reflect on them'.*

## 5. Conclusions

University museums provide unique opportunities to actively fulfil critical mission statements
of higher education institutions (1; 2; 3). Thus, it is necessary to think about efficient strategies to
exploit University cultural heritage resources. Research has highlighted the role of digital technologies
to improve users' experience within the museum context. In this article, we presented a case study
carried out at the Department of Educational Sciences of Roma Tre University which hosts a permanent
exhibition of the contemporary artist 'Tito Rossini'. The exhibition was equipped with Augmented
Reality Technology, designed through a simple gamification *concept*. Students that took part in the
experimentation expressed an overall positive evaluation.

Most of the participants did not know about the exhibition neither the artist before the visit. This
means that the Department of Education should find efficient solutions to better exploit its cultural
heritage resources.

The exhibition seemed to have an impact on students' critical thinking enhancement because
participants often reported topics related to thinking processes such as *inquisitiveness* and *analysis*.
The last question of our questionnaire was aimed to assess participants critical thinking levels. More
specifically, the question was designed to stimulate students to reflect upon the role of university
museums in the broader society. Most of the participants expressed critical reflections in their

answers and they obtained scores higher than 18/30 which corresponds to sufficiency in University Italian systems

Technology was considered both a strength and a weak point. For the next experimentation in the program, we will try to combine previously realized techniques with digital storytelling methodology, inserting stories and interactive music within the experience and giving users the chance to participate in the project by creating their digital content.

Although the limited number of participants (14 students) does not allow any generalisation of the results, this pilot study provides initial highlights regarding the positive impact of the use of AR in enhancing the fruition of a university museum collection.

**Author Contributions:** Conceptualization, A.P. and M.R.R.; methodology, F.A.; software, C.D.M.; validation, C.D.M. and M.V.; formal analysis, F.A.; investigation, C.D.M. and M.V.; resources, C.D.M. and M.V.; data curation, F.A.; writing—original draft preparation, F.A.; writing—review and editing, A.P.; visualization, F.A.; supervision, A.P.; project administration, A.P.

**Funding:** This research received no external funding.

**Conflicts of Interest:** The authors declare no conflict of interest

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
