# Peer review of "Adopting Augmented Reality to Engage Higher Education Students in a Museum University Collection: the Experience at Roma Tre University"

_information, doi:10.3390/info10120373_

Round 1

Reviewer 1 Report

They should include a conclusion section.
They should improve the review and theoretical foundation of the work with references to more current and research papers.

Author Response

Dear reviewer, 

as required, we implemented the conclusion section and the review and theoretical foundation of the work with references to more current and research papers.

Thank you for your comments. 

The authors

Reviewer 2 Report

The paper entitled "Adopting Augmented Reality to engage Higher Education students in a Museum University Collection: the experience at Roma Tre University" contains observations of the effects of introducing augmented reality means into presenting museum exhibitions to the visitors, in this instance - university master students. The study relates both to the design of the interactive software and to the personal experience of its introduction to the users. Original results are reported from which directions for future software enhancements are derived as well as some recommendations for more active promotion of cultural heritage among the younger population. The paper is well written in clear technical English, with the exception of a few grammar inaccuracies listed below, which would be easy for reading by the general audience.
Some complements related to the substance of the presented approach would make the manuscript better, that is:
- What is the effect of allowing the user to answer a question unlimited nuber of times on his/her overall performance in the context of the aims of the study and proposed methodology of teaching inducing critical thinking? Some observations towards this direction would be proper for the authors to share with the reader. - Why part of the final question was shown each time the user manage to earn a point but not the whole one at last - some motivation on that functionality could be included in the expose.

Remarks related to the content of the study:
- What is F = 12 and Other = 2 on line 127? These parameters should be declared explixitly in the text.
- What are 'percentage frequencies' mentioned at the end of Section 3? Some short, perhaps informal, definition should be added in the text and/or the term should be precised, probably substituted with some more common one.
- A reference to the Atlas.ti8 software is preferable to exist in the text.
- By what criteria a score of 18/30 is considered as satisfactory when evaluating the level of Critical Thinking (line 275) - a reference is needed.
- Section '5. Discussion' is actually a Conclusion considering its current content. The paper may very well be extended with a detailed dicussion on experimental results if the authors decide to do so.
- 'Author Contributions' section is empty.

Some minor technical remarks:
- The abbreviation AR in the Abstract should be substitued with its full expression;
- The paragraph from lines 57-60 should be placed at the end of Section 1;
- On line 115, the "a first phase" hould be "the first phase";
- On line 124, "during" should be removed;
- On line 127, the sentence should not start with a number;
- The sentence on lines 129-133 should be split in 2 or more simpler sentences;
- On line 141, "artistic knowledge" should be "knowledge on art";
- In Table 1, "Blac" color probably should be "Black"; The title of the same table is half in Bold, only the "Table 1" part should be and the same is with the titles of Figures 2-6;
- On line 184, "humanoid shaped, which can identified" should be "humanoid shape, which can be identified";
- On line 186, "camera's mobile device" should be "mobile device camera";
- On line 190, "wants" should be "want";
- On line 191, "disappoint" should be "disappointment";
- On line 197, "file rouge" should be "file rouge";
- On line 209, "assess" should be "assessment of";
- On line 214, "method" should be "methods";
- On line 215, "trhough" should be "through";
- On line 243, "for the improvement for the next experimentation" should be "for improvement of the next experimentation";
- On line 249, "there were emerged the following topics" should be "the following topics emerged";
- On line 265, "considered both" should be "considered in both";
- On line 266, "upload" should be "install";
- Figure 6 should be explicitly referenced prior to its appearance in the text. How the maximum value of 30 for the Critical Thinking level was set up (to what it corresponds)?
- On line 293, the sentence should not start with a number.

I recommend the paper to be accepted after minor revision considering the remarks from above.

Author Response

Dear reviewer, 

thank you for your helpful and detailed comments, they were helpful to implement the quality of our paper.

We accepted all the comments you made regarding the English Language and style. In addition, we made the following implementations:

We add an explanation of the reason why we decided to allow participants to answer each question an unlimited number of times:

This because the questions of the quiz were not aimed to provide a summative assessment for the teacher but to enhance participants’ interaction with the painting and engagement during the visit.

We substitute the term frequencies with the term occurrence, easier to understand and we provide a better explanation of what we mean;

We explained why a score of 18/30 is considered as satisfactory when evaluating the level of Critical Thinking. The reference provided is the Italian University system assessment.

We implemented the conclusions. 

We hope you considered satisfactory the updated version of our manuscript.

The authors

Reviewer 3 Report

The paper has serious flaws as highlighted below:

1. The language is very problematic.

2. The sample size (N=14) is too low.

3. The supplementary data such as student names and their responses would have been provided.

3. The 'Literature review' section is missing.

4. The 'Conclusions' section is missing.

Author Response

Dear reviewer,

thank you for your comments. We tried to use them to implement the quality of our paper and we made the following implementation:

- We are aware of the limitation regarding the sample size, but we proposed this experimentation only as a pilot study. We add a more detailed explanation in the conclusion section:

Although the limited number of participants (14 students) does not allow any generalization of the results, this pilot study provides initial highlights regarding the positive impact of the use of AR in enhancing the fruition of a university museum collection.

- We did not collect students’ names and we explained why.

The questionnaire was filled in anonymously in order to respect the participants’ privacy and not inhibit participants to also providing negative feedback regarding the exhibition.

Eventually, we implemented both the literature review (in the paragraph introduction) and the conclusion section.

In addition, our manuscript was further checked by professional English editing service.

We hope our implementation could be considered satisfactory for you.

Authors

Round 2

Reviewer 1 Report

The work has improved.

Author Response

Thank you!

Reviewer 3 Report

I've now reviewed your revised paper and the replies to my comments. Though as authors said it is a pilot study, the low sample size can not be accepted in any case whether it is pilot or complete study. A good research study needs to be based on solid and agreed grounds. The authors also failed to provide satisfactory answer to my questions on participants' demographics. I'm not convinced. 

Author Response

Dear reviewer, 

according to your requirement, we summarized the demographic data collected in table 2 of the manuscript. 

This AR fruition in the university museum collection was tested for the first time and we needed to involve a small group of participants to monitor and support them (we inserted this explanation in the manuscript).  

Although the results cannot be generalized, in our perspective this pilot provides some interesting highlights regarding the impact of AR in the small group of participants involved and it opens up reflections for future research.

We hope you can consider the updated version of our paper.

Best,

The authors.